# Imaging breast malignancies with the Twente Photoacoustic Mammoscope 2

S. M. Schoustra[1,2], B. De Santi[1], T. J. P. M. op 't Root[3], C. A. H. Klazen[4], M. van der Schaaf[4], J. Veltman[1,5], W. Steenbergen[2], S. Manohar[1]*

1 Multi-Modality Medical Imaging (M3I), Technical Medical Centre, University of Twente, Enschede, The Netherlands, 2 Biomedical Photonic Imaging Group, Technical Medical Centre, University of Twente, Enschede, The Netherlands, 3 PA Imaging R&D B.V., Enschede, The Netherlands, 4 Medisch Spectrum Twente, Enschede, The Netherlands, 5 Ziekenhuisgroep Twente, Hengelo, The Netherlands

* s.manohar@utwente.nl

**Data Availability Statement:** Data has been uploaded to the figshare repository: https://figshare.com/s/030561cd4ca6f1c83256.

## Abstract

Clinical measurements on breast cancer patients were performed with a three-dimensional tomographic photoacoustic prototype imager (PAM 2). Patients with a suspicious lesion, visiting the center for breast care of a local hospital, were included in the study. The acquired photoacoustic images were compared to conventional clinical images. Of 30 scanned patients, 19 were diagnosed with one or more malignancies, of which a subset of four patients was selected for detailed analysis. Reconstructed images were processed to enhance image quality and the visibility of blood vessels. Processed photoacoustic images were compared to contrast-enhanced magnetic resonance images where available, which aided in localizing the expected tumoral region. In two cases, spotty high-intensity photoacoustic signals could be seen in the tumoral region, attributable to the tumor. One of these cases also displayed a relatively high image entropy at the tumor site, likely related to the chaotic vascular networks associated with malignancies. For the other two cases, it was not possible to identify features indicative of malignancy, because of limitations in the illumination scheme and difficulties in locating the region of interest in the photoacoustic image.

## Introduction

Photoacoustic imaging is being thoroughly researched for its potential role in, among many other areas, breast cancer management [1, 2]. Breast cancer is the cancer type in females with the highest number of newly diagnosed cases each year [3]. The current clinical imaging methods for detection, diagnosis and evaluation of breast cancer have advantages, but also drawbacks. Photoacoustics is a noninvasive functional imaging method that does not use ionizing radiation, a contrast agent nor apply breast compression. The breast is illuminated with short pulses of light. Selective absorption of the light takes place in the tissue, and thermal expansion in turn generates ultrasound waves that can be detected outside the body and reconstructed into an image. When choosing a wavelength of light which is strongly absorbed by hemoglobin, we obtain a map of the vasculature, which is of interest given the relationship between

**Funding:** Stichting Achmea Gezondheidszorg, Project No. Z620., author W.S., https://www.zilverenkruis.nl/zorgaanbieders/visie-en-beleid/innovatie/sag Health~Holland LSI-TKH PPP project SACAMIR (No. LSHM17007), author S.M., https://www.health-holland.com/funding-opportunities/tki-match PAMMOTH project funded by the EU's Horizon 2020 RIA H2020 ICT 2016-2017 under Grant Agreement No. 732411, author S.M., https://ec.europa.eu/info/research-and-innovation/funding/funding-opportunities/funding-programmes-and-open-calls/horizon-2020_en The funders had no role in study design, data collection and analysis, decision to publish, or preparation of the manuscript.

**Competing interests:** Co-author T. J. P. M. o. R. is employed by PA Imaging R&D B.V but has no financial interest in the company. Co-author C. K. has a financial interest in PA Imaging Holding B.V., via PAMARA Holding B.V. This does not alter our adherence to PLOS ONE policies on sharing data and materials. The authors declare that there are no other conflicts of interest.

vascularization and cancer progression [4]. The method could potentially have a role in the area of breast cancer management, if it proves to have an added value over current methods.

Three-dimensional photoacoustic imaging of breast cancer has been reported earlier [5–10]. Toi et al. scanned 22 patients with malignant breast tumors with their PAM-03 system, employing a hemispherical detector array with 512 elements [7]. The breast was illuminated with two wavelengths: 755 and 795 nm. PA images were analyzed as well as fused MR-PA images. In the latter, MR an PA images were registered after MR images were deformed to match sizes and orientations in PA images using common landmarks in both modalities. From manual counting of trunks and branches of blood vessels after elimination of subcutaneous vessels, it was concluded that often, there were more PA signals considered to be of blood vessels in the affected breast than in the contralateral breast. Differences between PA appearance of ductal carcinoma in situ cases (DCIS) and invasive breast cancers were studied. Significantly more centripetal blood vessel structures were found in invasive breast cancers compared to the DCIS cases. Many invasive breast cancer cases showed vessel disruption or rapid narrowing at the boundary of the tumor. In one case, changes in intratumoral blood vessels were observed after chemotherapy. Hypoxic spotty signals could be seen in the tumor, not present before therapy. After upgrading the system to PAI-04 [8], with higher frequency detectors and the co-acquisition of ultrasonic images among other advances, another breast tumor was imaged. The three-dimensional US image aided in identifying the tumor position. Fine, tumor-related blood vessels could be visualized as well as scattered hypoxic points, hypothesized to be attributable to intratumoral bleeding.

Lin et al. detected all nine tumors (of which six were invasive carcinomas) in seven patients with their single-breath-hold photoacoustic tomography (SBH-PACT) system [9]. The entire breast was illuminated with 1064 nm and scanned within ~15 seconds. Eight out of nine tumors were detected at the expected location by observing higher blood vessel densities. A vessel density map of the breast was calculated and a threshold was set to distinguish high vessel densities from normal vessel densities. The one tumor that was not detected by looking at abnormal vessel densities, was identified by investigation of the compliance of the breast tissue (photoacoustic elastography). A difference in compliance between the tumoral region and surrounding normal tissue indicated the ninth tumor. The same system was used to evaluate response to neoadjuvant chemotherapy (NAC) [10]. Three breast cancer patients receiving NAC treatment were scanned before, during and after treatment. Again, vessel density maps were calculated to distinguish the tumor region from healthy tissue. Moreover, entropy and anisotropy were evaluated. Binarized anisotropy-weighted entropy maps were employed to acquire cancer masks, which in turn were used to calculate and evaluate tumor sizes.

Heijblom et al. imaged 33 malignancies with the first generation Twente Photoacoustic Mammoscope (PAM) system, employing a planar configuration where the breast was mildly compressed in a craniocaudal direction [5, 6]. In 32 out of 33 cases, the malignancy could be identified through PA imaging. Photoacoustic lesion appearance was studied and three main types were observed: 'mass', 'non-mass' and 'ring'. By comparing PA images not only to MR images but also to vascular stained histopathology (for some cases), PA intensities were attributed to the presence of vascularity. Continuing research on photoacoustic mammography, the second generation Twente Photoacoustic Mammoscope (PAM 2) was developed, where some of the limitations of PAM 1 were overcome. PAM 1 had a limited field of view ($90 \times 80 \text{ mm}^2$ detector surface area), made use of breast compression and the flat 2D detector array complicated 3D visualization of lesions. In PAM 2 the breast is pendant in water without any compression, with illumination and ultrasound detection from around the breast. Illumination is performed with 10 beams (as opposed to one large beam in PAM 1), where one beam is directed at the ventral side of the breast, and nine smaller beams directed at the region of the

breast close to the chest wall. Further, illumination can be provided at 2 wavelengths 755 and 1064 nm (as opposed to single-wavelength illumination at 1064 nm in PAM 1). The ultrasound detectors are arranged as 12 arcs with 32 elements each around the breast; the arc-shapes follow the pendant breast contour. Over the course of 1.5 years, we performed a clinical study with our prototype tomographic PAM 2 system for breast imaging [11, 12]. Thirty patients were measured, with benign as well as malignant abnormalities (diagnosed by conventional imaging modalities and/or histopathologic examination). The aim of this study was to investigate or find possible markers indicative of malignancy. In this paper, we show in depth results of four case studies, all breast cancer patients with one or more malignant breast lesion. Reconstructed images are processed and compared to contrast-enhanced MR images if available. Per case, a tailormade image analysis was done and observations of the photoacoustic appearance of the tumoral region are presented.

## Materials and methods

### Study design and clinical setting

Photoacoustic measurements were performed on patients visiting the center for breast care of the Medisch Spectrum Twente hospital (Oldenzaal, The Netherlands). Patients with a lesion suspicious for malignancy, classified as BI-RADS 4 or 5 through clinical investigation and conventional diagnostic imaging, were asked to participate in our study. Also, patients who came to the hospital with a suspicious lesion which was classified as BI-RADS 2 or 3 after clinical investigation and conventional diagnostic imaging were asked to participate. All subjects were adults and fully competent to give informed consent. Exclusion criteria were: a breast biopsy in the six months prior to this study; bloody discharge, ulcers or wounds on the breast; a history of surgery (including cosmetic surgery) or radiation therapy on the breast; chemotherapy at the time of this study. A cup size of D or larger was an exclusion criterion. Approval for the study protocol and procedures was obtained from an institutional review board (METC Twente, Medisch Spectrum Twente, Enschede, The Netherlands) and the study was registered in the Netherlands National Trial Register (NL6010). Written informed consent was obtained from all participating patients.

Upon arrival at the hospital, each patient had a physical examination and anamnesis by a nurse practitioner. Then, conventional images were obtained (x-ray and/or US). For patients referred through the screening program, the x-ray screening images were examined and in some case additional x-ray imaging was performed. Craniocaudal (CC) and mediolateral-oblique (MLO) tomosynthesis x-ray images of both breasts were obtained with a Hologic Dimensions 3D Mammography system (Hologic, Inc., Marlborough, Massachusetts). US imaging was done with a Philips EPIQ 5G system (Koninklijke Philips N.V., Amsterdam, The Netherlands), with the L12-5 50 mm linear array transducer and/or the L18-5 broadband linear array transducer, depending on tissue and lesion characteristics and radiologist's experience. If somewhere in this process, a patient met all inclusion criteria and was willing to participate, PAM 2 scans of both breasts were made. This was always done after or in between regular imaging procedures, and always before a US-guided core needle biopsy (if applicable). A subset of patients was scheduled for MR imaging, when additional imaging information was needed for diagnosis and/or treatment planning. For this, a Philips Ingenia 3.0T system (Koninklijke Philips N.V., Amsterdam, The Netherlands) was used with a dStream breast 16ch coil. High-resolution, anatomic T2-weighted images and dynamic T1-weighted images were obtained before and after administration of gadolinium contrast medium. All conventional images were interpreted by a breast radiologist and biopsy specimens were examined by a pathologist. Radiologists' and histopathologists' reports were

made available to the researchers. Final diagnoses were either based on US (in the case of cysts) or histopathological examination.

## PAM 2 system

The PAM 2 system was first described in Ref. [11] and is shown in Fig 1. The tomographic system images one breast at a time, inside an imaging tank filled with water while the subject lies prone on a bed. The pendant breast is illuminated by a dual-head laser (755 and 1064 nm) by multiple beams: one directed at the nipple (50% of the energy) and nine directed at the sides (50% of the energy is spread over these nine fibers bundles) of the breast. US signals are detected with 12 arc-shaped arrays, each containing 32 piezocomposite elements (Imasonic SAS, Voray-sur-l'Ognon, France) with a center frequency of 1 MHz. The imaging tank, including illumination bundles and detector arrays, rotates around the breast in steps. In this way, 45 projection angles (total covered angle of 60 deg) are acquired and in each position, signals of 35 pulses of each wavelength are averaged. Obtaining a scan of one breast following this protocol took approximately four minutes. The breast containing a lesion was scanned as well as the contralateral breast. Subjects were scanned with one breast freely pendant in the water inside the imaging tank. For reconstruction of images, a filtered backprojection algorithm was employed using averaged signals, as described in Ref. [11]. The algorithm was supplemented with an iterative scheme (five iterations), as described in Ref. [12]. Images were reconstructed with a homogeneous speed of sound (SOS) based on the coupling water temperature [13],

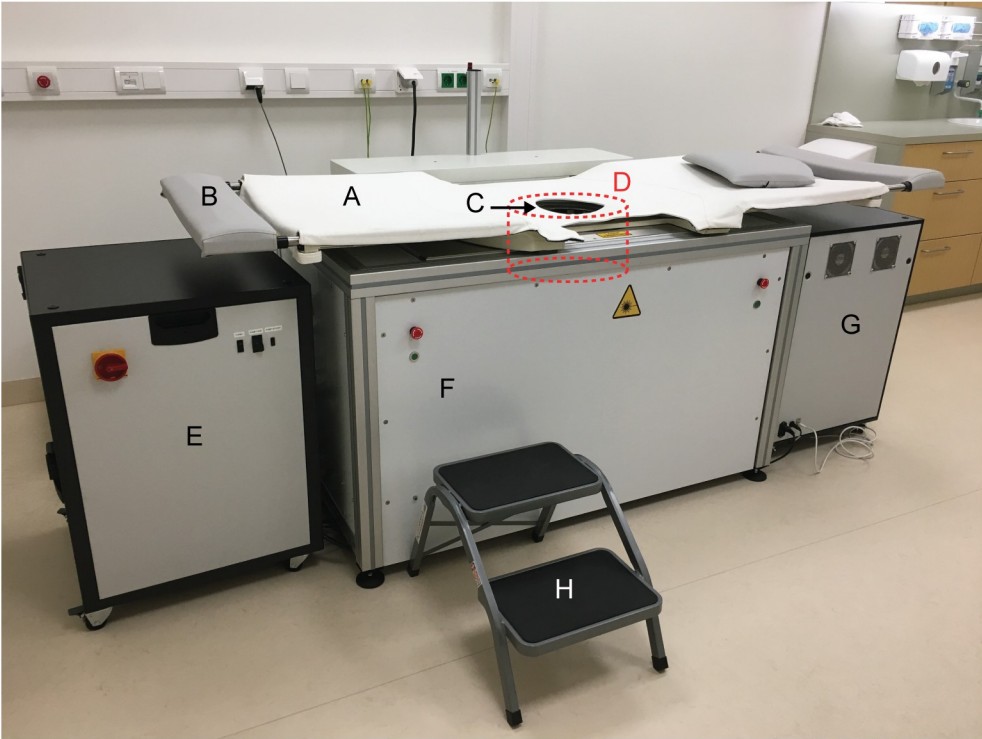

**Fig 1. PAM 2 system in hospital setting with (A) the bed on which a woman lies prone, (B) a foot rest, and (C) the breast aperture with (D) a schematic of the shape and position of the imaging tank below the bed, (E) the power supply and cooling unit, (F) laser head (behind the panel), (G) DAQ unit, and (h) step stool.** Reproduced from Ref. [11] with permission of authors and publisher.

which was measured during each scan. The resolution of all reconstructed volumes (voxel size) is 0.4 mm in each direction.

## Image processing and interpretation

A subset of measurements was excluded from further analysis based on quality assessment. The absence of the visibility of a breast contour and/or nipple, as well as the presence of smears indicative of blurring due to motion artifacts were indications of unacceptable quality. Quite often, very few or even no discernable blood vessels were reconstructed in these images. Secondly, based on information available from the clinic, the visibility of the region(s) of interest (ROI), containing one or more malignancies, was investigated for each measurement. In cases where MR images were available, these were compared to PA images to identify landmarks and locate the expected lesion location. Measurements for which it was concluded that the ROI was not imaged properly with PAM 2 were excluded from further analysis.

From the remaining subset, four subjects with a malignant lesion were chosen for further analysis. Reconstructed scans of these subjects were processed mainly for two reasons: (i) improving the visualization of blood vessels; and (ii) assessing the ability of quantitative image biomarkers to distinguish tumoral tissue from healthy tissue in the images.

A processing pipeline was adopted for every selected subject All processing steps were done with MATLAB (R2021b, The MathWorks, Inc., Natick, Massachusetts). Fig 2 shows the results of all steps for one of the scanned patients (case 1). The 755 nm and 1064 nm reconstructed PA volumes were loaded and intensity values were scaled from 0 to 1 (Fig 2A and 2B). Median filtering with a moving kernel of $3 \times 3 \times 3$ voxels was performed in order to reduce noise (Fig 2C and 2D). Frangi vesselness filtering [14] was applied in order to enhance the contrast of blood vessels in the images (Fig 2E and 2F). To further improve image contrast, adaptive intensity modulation was applied [15]. In this, the Frangi vesselness filtered images were modulated by dividing them by standard deviation maps. During this process, the image was split into $11 \times 11 \times 11$ voxel patches. The intensity of the voxels was normalized to the standard deviation of the intensities within the patch. Hereby, portions of the image with low variability were enhanced in order to bring out structures that would otherwise not be visible in the image,

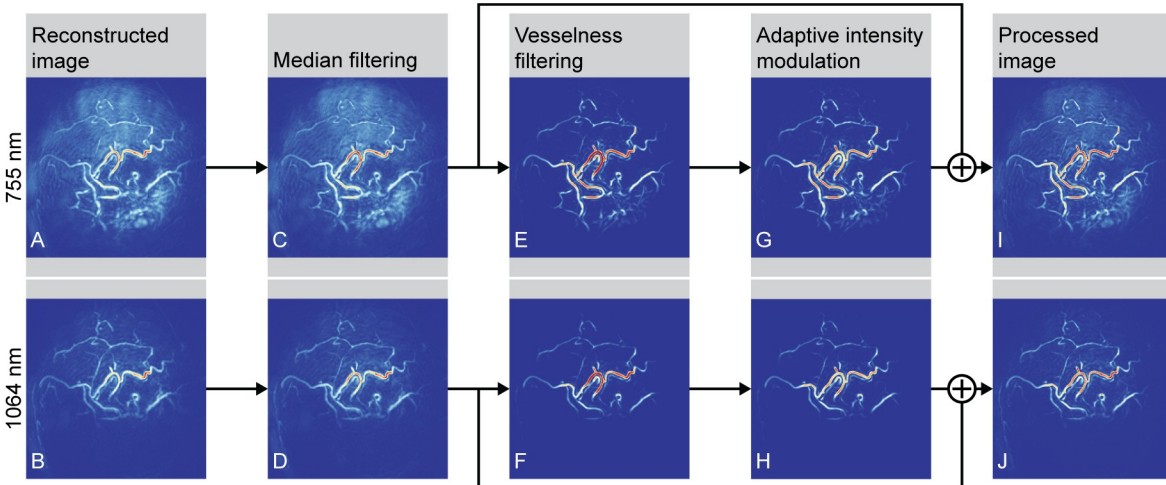

**Fig 2. Processing pipeline.** Coronal view maximum intensity projections of case 1 are shown, to illustrate the different processing steps. Top row shows images obtained by illuminating the left breast of the patient with 755 nm, bottom row shows the same breast illuminated with 1064 nm.

while portions with high variability in intensity were attenuated. As a main result, this improved images in terms of homogeneity of the voxel intensities, mitigating the problem of non-homogeneous illumination. The final processed images were obtained by summing the modulated image with the median filtered images (Fig 2G and 2H).

If available, MR images were studied and compared to PA images. PA images were transformed (rotation and axes flipping) to match the conventional image orientation system of MR images. Maximum intensity projections (MIPs) of tissue slabs with varying thicknesses (from 5 to 20 mm) were computed for PA as well as MR images, to make the image comparison easier. By scrolling through these slabs, similarities between vascular structures in MR and PA were sought, which was helpful in locating the tumoral region.

Next to the processing pipeline as described above, we also calculated entropy maps. They statistically describe the degree of randomness of the distribution of PA amplitudes in local regions of the image. Since angiogenesis induces the formation of chaotic vascular networks, one of the expected image features would be the presence of vessels showing abrupt changes in amplitude [10]. First, the local entropy was calculated. A cubical moving window of $4 \times 4 \times 4$ mm was used to scan the whole volume and obtain a local entropy map. Due to rapid transitions between background tissue and blood vessels, also healthy structure can present high local entropy values. Therefore, the obtained entropy map was corrected using the response of a vesselness filter [14]. The local entropy map was then corrected with the vesselness filter response, to obtain the final entropy map. In this study, we report the entropy maps only for one case (case 3), which showed relatively good illumination in the region of the tumoral and nearby regions. For other cases, this was not the case, complicating interpretation of an entropy map.

## Results

From March 2018 until November 2018, 33 patients were included in the study. See Table 1 for an overview. Of this group, three women were not measured: two subjects were unable to climb and/or lie prone on the imaging bed, and one subject could not be measured due to technical issues with the system. In general, both breasts were scanned. For two patients, one of the breasts was not scanned because of a history of surgery or amputation of one breast. Out of 30 measured patients, 19 were finally diagnosed with a malignant abnormality (M in Table 1). In this work, we focus on investigation of PAM 2 appearance of malignancies. Of these 19 patients, a selection of four was made to be thoroughly analyzed in this paper. The 15 patients with diagnoses of malignancies that were not included in this selection, were studied outside the scope of this work, where photoacoustic images of these patients were found to either: (i) be of such low quality that it was hard or even impossible to discern the boundary of the breast, the nipple, and/or vasculature; or (ii) contain smears or blurring, due to motion artifacts; or (iii) not have optimal positioning of the region of interest (ROI) containing the malignancy (as deduced from conventional imaging and/or clinical descriptions).

S1 Table shows an overview of the characteristics of all 30 measured subjects. The average age of the 30 subjects was 53, of which the youngest was 30 and the oldest 79 years of age at the

**Table 1. Study overview.**

| First patient | Last patient | Included | Measured | Dx | |
|---|---|---|---|---|---|
| | | | | B | M |
| March 2018 | November 2018 | 33 | 30 | 11 | 19 |

'Dx' = final diagnosis, 'B' = benign lesion, 'M' = malignant lesion.

**Table 2. Characteristics of the four breast cancer patients and subjects, studied in detail.**

| Case | Estimated size of lesion (mm) | Diagnosis biopsy | Treatment | Histopathology after treatment |
|------|-------------------------------|------------------|-----------|-------------------------------|
| 1 | 19 | Invasive carcinoma NST grade 1 DD tubular carcinoma, multicentric | Mastectomy | Invasive carcinoma NST grade 1 |
| 2 | 39 | Invasive carcinoma NST grade 3 | Mastectomy | Invasive carcinoma NST grade 3 |
| 3 | 33 | Intraductal papillary carcinoma (uncertain) | Mastectomy | Mucinous carcinoma |
| 4 | 25 | Invasive carcinoma NST grade 3 | NAC + lumpectomy | Invasive carcinoma NST grade 2 |

'NST' = no specific type, 'DD' = differential diagnosis, 'NAC' = neoadjuvant chemotherapy.

time of inclusion. Further, S1 Table shows the diagnosis of the suspect lesion, location and size, and the availability of MR images. Estimated lesion size was measured on MR images when available, otherwise it was estimated from x-ray and/or US images. When the lesion size was measured in three dimensions, the largest dimension is reported. In cases where satellite lesions were detected through MR, the reported size refers to the main lesion. Final diagnoses were either based on US (in the case of cysts) or histopathological examination, which was performed on excised (biopsied) tissue and/or specimens obtained from a lumpectomy or mastectomy. The latter is only mentioned for the four selected patients in Table 2, which summarizes characteristics of these four subjects.

## Case studies

**Case 1.** Patient 1 (58 years old) was referred to the hospital through the national screening program. X-ray images showed dense structures, with most dense tissue in upper outer quadrants of both breasts. Centrally located in the upper part of the left breast, architectural distortion with a spiculated mass was observed, which was classified as suspicious (BI-RADS 5) (Fig 3A). US investigation (Fig 3B) revealed a suspect mass with a diameter of 12 mm, at 11 or 12 o'clock in the left breast (L1). A smaller lesion (4 mm) was located slightly more ventrally in the breast (L2), lateral to the other mass. Three 14G biopsies were taken from the largest mass. Histopathological examination of the specimens revealed an invasive carcinoma no special type (NST) or a tubular carcinoma (differential diagnosis), Bloom and Richardson grade 1. Additional MR investigation (Fig 3C) showed a total of five suspicious lesions (of which one was the pathology-proven carcinoma), located centrally and medially in the left breast. Based on this, a multifocal carcinoma in the left breast was suspected. Histopathological examination after mastectomy revealed two tumor foci, both invasive carcinomas NST of grade 1.

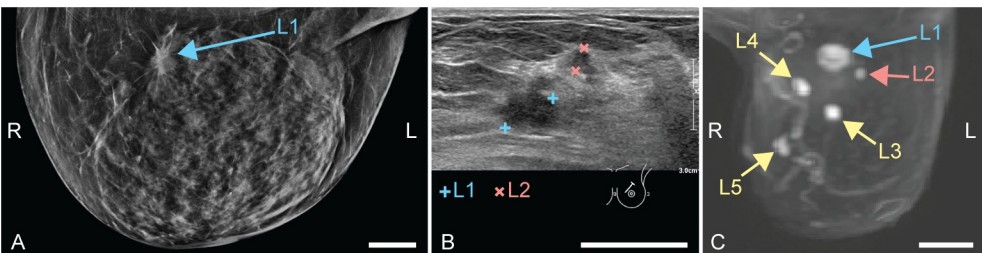

**Fig 3. Case 1: Diagnostic images of a multifocal carcinoma in the left breast.** L and R indicate left and right, respectively. (A) Craniocaudal x-ray (CC-MMG), with the suspect mass indicated by L1. (B) Ultrasound (US) image, where the main lesion is indicated with blue and a satellite lesion with pink. (C) Perfusion maximum intensity projection magnetic resonance (MR) image. The initially detected mass is indicated by L1. The satellite lesion as detected by US is L2. Three more foci were found though MR investigation: L3, L4 and L5. All scale bars represent 20 mm.

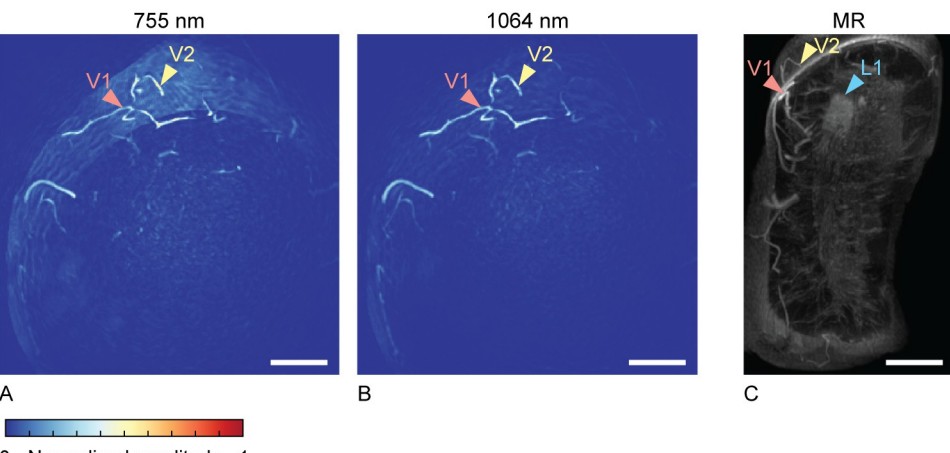

**Fig 4. Case 1: Local MIPs of slabs with a thickness of 20 mm, near the main lesion, close to the chest wall, in the coronal view.** Photoacoustic images at 755 nm (A) and 1064 nm (B), compared to a post-contrast dynamic T1 MR MIP image (C). All scale bars represent 20 mm.

Photoacoustic reconstructions were processed through the steps described in Section 2.3. Fig 4A and 4B show processed photoacoustic MIPs of slabs with a thickness of 20 mm in the coronal plane, close to the chest wall. It was attempted to show the same slab of tissue in Fig 4A and 4B as in Fig 4C, where we can identify one of the tumor foci (L1). Fig 4C shows a MIP of a slab of 20 mm of a post-contrast MR image. L1 points at one of the five tumor foci from Fig 3C. The vascular structure indicated with V2 in Fig 4 has a similar appearance in both modalities and can therefore serve as a landmark. V1 points at another vessel, identifiable in all 3 images. The photoacoustic images do not show anything remarkable at the expected tumor location. However, it is difficult to exactly pinpoint the expected location, given the lack of recognizable landmarks nor anatomical overlap between the two modalities.

Photoacoustic reconstructions at both wavelengths, in the coronal and transverse plane, are presented and compared to post-contrast dynamic T1 MR MIPs in the same two planes in S1 Fig.

**Case 2.** Patient 2 (79 years old) presented with a suspicious mass of 25 mm in the lower inner part of the right breast, located right under the skin. Local skin retraction was observed. X-ray imaging showed a mass at 6 o'clock, situated against the thoracic wall (Fig 5A). Part of the lesion was outside the field of view, making it difficult to assess lesion dimensions. The

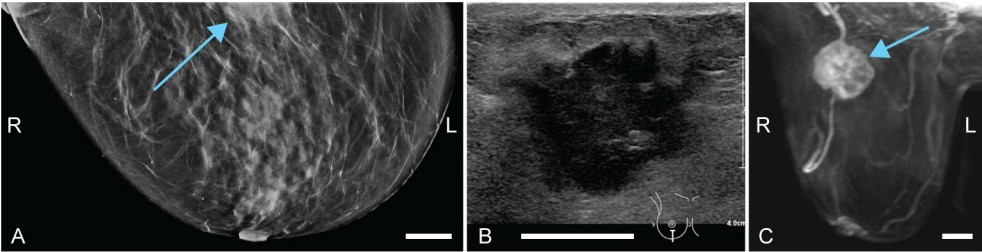

**Fig 5. Case 2: Diagnostic images of an invasive carcinoma in the right breast.** L and R indicate left and right, respectively. (A) Craniocaudal x-ray (CC-MMG). (B) Ultrasound (US) image, where the lesion is clearly visible (C) Perfusion maximum intensity projection magnetic resonance (MR) image. All scale bars represent 20 mm.

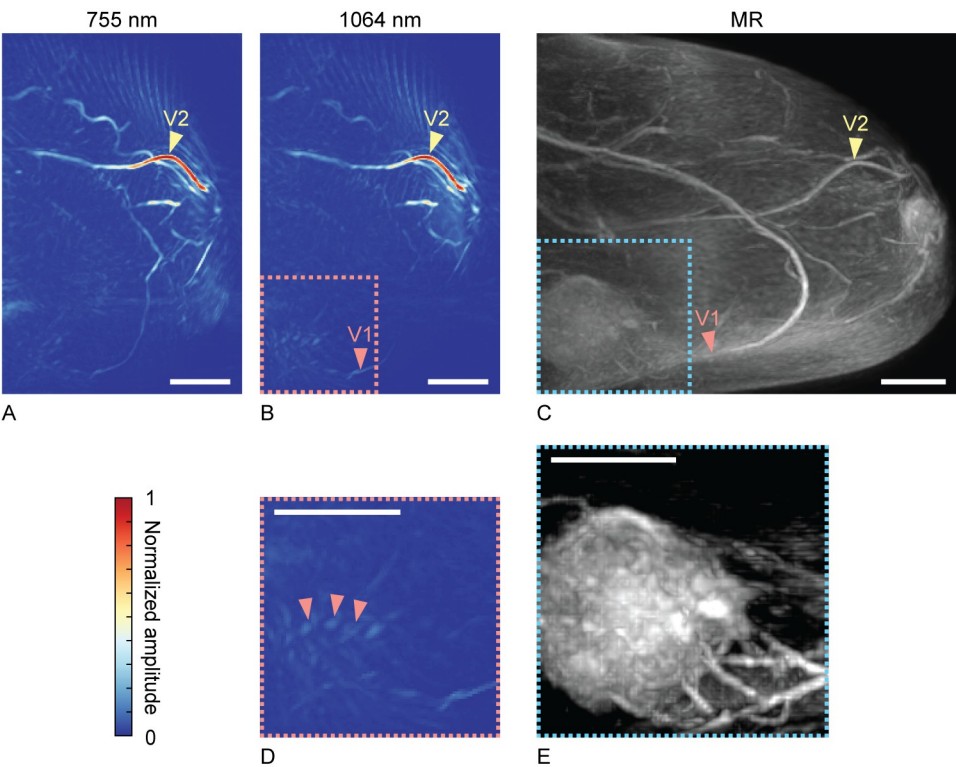

**Fig 6. Case 2.** Photoacoustic MIPs at 755 nm (A) and 1064 nm (B) in the sagittal plane. (C) Post-contrast dynamic T1 MR MIP in the sagittal plane. (D) Zoomed subsection of (B). (E) Zoomed subsection of the region marked in (C), where a MIP of a slab of tissue with thickness 50 mm is taken (containing the entire tumor); contrary to (C), where a projection of the entire breast is taken. All scale bars represent 20 mm.

lesion presented on US as a lobed mass, surrounded by hyperechogenic tissue (Fig 5B). The estimated size was around 30 mm. Two 14G biopsies were taken from the BI-RADS 5 mass, which were indicative of a grade 3 invasive carcinoma NST upon histopathological investigation. The same mass was identified on MR images (Fig 5C) and was measured to be 36 × 25 × 35 mm. There was no ingrowth of the lesion into the pectoralis muscle. No other (new or satellite) lesions were found through MR investigation. The patient had a mastectomy of the right breast. Histopathological examination of the lesion in the removed breast confirmed the diagnosis of an invasive carcinoma NST grade 3.

Photoacoustic images were processed via the method described in Section 2.3. Fig 6A and 6B show processed MIPs in the sagittal plane. The post-contrast dynamic T1 MR MIP in Fig 6C shows similar structures. V1 indicates a large blood vessel going towards the tumor, visible in all images and with a similar shape. V2 indicates a prominent vessel with a recognizable bulge. The tumor is well visible on the post-contrast MR of Fig 6C, in the dashed box in the inferior part of the breast. This part of the breast is shown enlarged in Fig 6E, where a projection of a slab with thickness 50 mm was made, containing the entire tumor. Window level and width values were adjusted to enhance contrast and visibility of the tumor and its surrounding vessels. A similar enlarged view of the same region of the PA MIP image at 1064 nm is shown in Fig 6D. The expected location of the tumor and thus the dashed box was deduced from structures V1 and V2. Arrowheads in Fig 6D point at spotty structures possibly attributable to the smaller tumor-related vessels or to higher intensity regions within the tumor, as seen in Fig

**Fig 7. Case 3: Diagnostic images of a mucinous carcinoma in the left breast.** L and R indicate left and right, respectively. (A) Craniocaudal x-ray (CC-MMG). (B) Ultrasound (US) image. All scale bars represent 20 mm.

6E. The latter is more likely, since the small vessels seem to be positioned in line with V1, and the spotty signals appear above that.

Photoacoustic reconstructions at both wavelengths, in the two other planes (coronal and transverse), are presented as MIPs and compared to post-contrast dynamic T1 MR MIPs in the same two planes in S2 Fig.

**Case 3.** Patient 3 (78 years old) self-detected a palpable lump in the left breast and was referred to the specialized breast clinic for further investigation. The abnormality had a size of approximately 30 mm and was located behind the nipple-areolar complex. X-ray images showed a sharply delineated lesion with embedded calcifications at the location of the palpable abnormality (Fig 7A). Targeted ultrasound (US) examination showed a lobed, sharply delineated, mostly solid but also partly cystic lesion at 9 o'clock in the areola (Fig 7B). In the center of the lesion a duct was visible which was connected to the nipple. The calcifications were visible with US as well. The lesion measured 33 mm on US. Intravascular vascularization was visualized by means of color Doppler. Three 14G core needle biopsy (CNB) specimens were taken from the most solid part of the lesion as identified on the US images as suspicious (BI-RADS 4). X-ray and US showed confusing features, leading to a broad differential diagnosis, including invasive ductal carcinoma, papillary tumor, phyllodes tumor, complex fibroadenoma and fibrocystic changes. Histopathological investigation of the biopsy specimens was suggestive of an intraductal papillary carcinoma, although atypical (BI-RADS 6). Mucinous carcinoma was also considered, but not all parts of the specimen were in line with that diagnosis. The pathologist was not certain based on the biopsy specimens. The patient had a surgery two weeks later. Histopathological investigation of the resected tumor revealed the tumor to be a mucinous carcinoma.

Photoacoustic images were again processed as described in Section 2.3. Fig 8A and 8C show processed MIPs of both wavelengths, in the transverse and sagittal plane respectively. For this patient, MR images were not available. However, the tumor location is apparent from physical examination and x-ray images: directly behind the nipple and very superficial. This region is indicated with a dotted ellipse in all views in Fig 8. In the PA images in the top row (Fig 8A and 8C), we can see many small spots with a relatively high intensity behind the nipple. Entropy maps corrected with a vesselness filter in the same views are depicted in Fig 8B and 8D. It can be seen that the region behind the nipple shows a relatively high level of entropy, best visible in the transverse plane (Fig 8B).

Photoacoustic reconstructions at both wavelengths, in the coronal and transverse plane, are presented as MIPs in S3 Fig.

**Case 4.** Patient 4 (56 years old) came to the hospital because of a self-detected lump in the left breast. In the lower outer quadrant of the left breast, a palpable abnormality of approximately

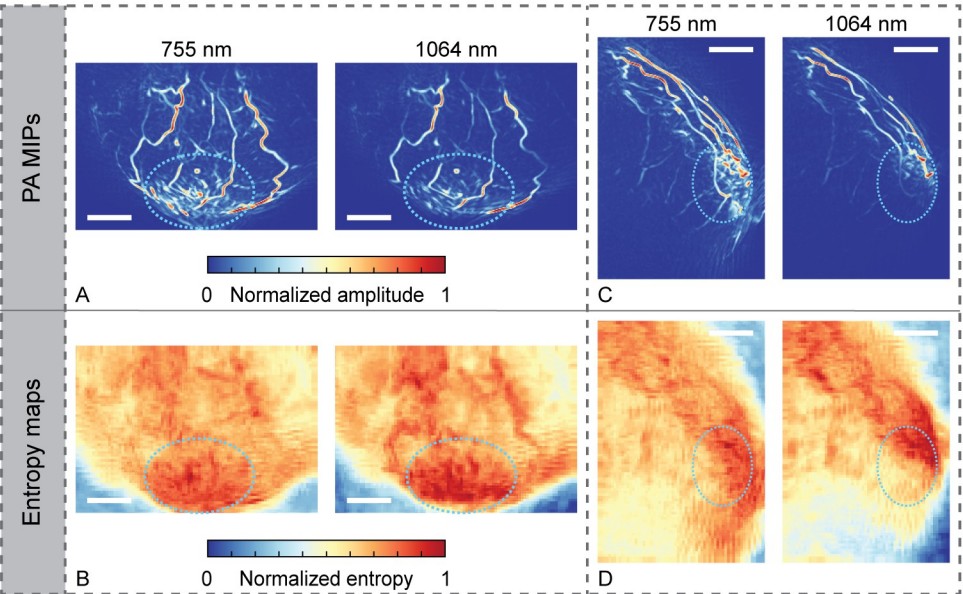

**Fig 8. Case 3: Photoacoustic maximum intensity projections (MIPs) and entropy maps in two planes, at two wavelengths.** (A)-(B) In the transverse plane. (C)-(D) In the sagittal plane. All scale bars represent 20 mm.

15 mm was noticed during physical examination. This presented as an unsharply delineated density of approximately 20 mm on the x-ray images (Fig 9A). US showed a suspicious looking hypo-echogenic lesion at the location of the palpable abnormality (Fig 9B). Two 14G biopsies were obtained from the BI-RADS 5 mass. The specimens taken from the breast were indicative of a grade 3 invasive carcinoma NST. In MR investigation (Fig 9C)), the lesion was identifiable with a few satellite lesions surrounding the main lesion. The total size of the region was 25 mm. The most dorsal lesion was 5 mm away from the pectoralis muscle. The distance to the skin was 8 mm. Histopathology after neoadjuvant chemotherapy and lumpectomy confirmed the diagnosis of an invasive carcinoma NST, but the material in this specimen was of grade 2.

Photoacoustic images were again processed as described in Section 2.3. Similar to case 1, processed photoacoustic coronal MIPs of slabs were obtained and studied, see Fig 10A and 10B. All slabs have a thickness of 15 mm and are located relatively close to the chest wall, where the tumor is situated. Fig 10C shows an MR MIP of a slab of tissue containing the tumor (L1). A vessel indicated by V1 is also visible in PA images of both wavelengths. Sprouting from the main vessel indicated by V1, we see multiple small vessels directed at the tumor

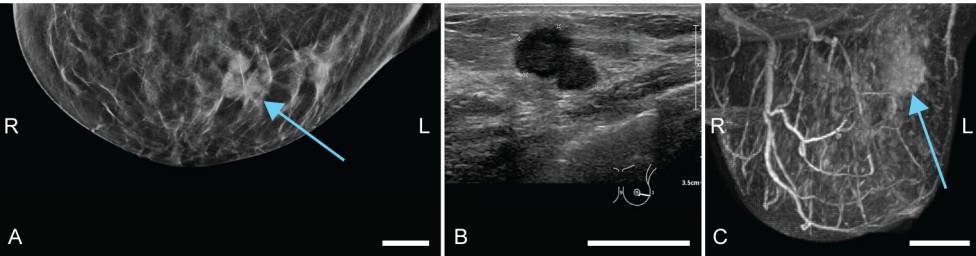

**Fig 9. Case 4: Diagnostic images of an invasive carcinoma in the left breast.** L and R indicate left and right, respectively. (A) Craniocaudal x-ray (CC-MMG). (B) Ultrasound (US) image. (C) MIP of the subtraction dynamic T1 MR acquisition, with adjusted window level and width values for enhanced contrast. All scale bars represent 20 mm.

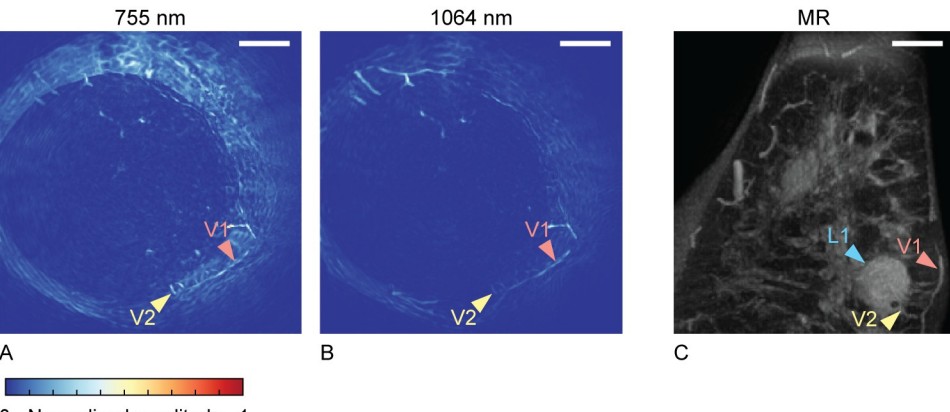

**Fig 10. Case 4: Local MIPs of slabs with a thickness of 15 mm, near the main lesion, close to the chest wall, in the coronal view.** Photoacoustic images at 755 nm (A) and 1064 nm (B), compared to a post-contrast dynamic T1 MR image (C). All scale bars represent 20 mm.

(V2 in Fig 10C), resembling a comb. These small vessels seem to also be apparent in PA images, best visible in the image obtained with 755 nm illumination (Fig 10A).

Photoacoustic reconstructions at both wavelengths, in the coronal and transverse plane, are presented as MIPs in S4 Fig.

## Discussion

In this work, we investigated 3D photoacoustic breast images, acquired with our PAM 2 system, of women diagnosed with one or more malignant breast lesions. A subset of four subjects was selected and presented. Reconstructed images were processed to enhance visibility of vascularization and decrease background noise. Processed images were analyzed and compared to conventional clinical images–x-ray, ultrasound and contrast-enhanced MR. The tumor site was localized in our PA images, after which an analysis of this region was performed. Given the visibility of vascularization in contrast-enhanced MR images and in our PA images, comparison with MR images was preferred, when available. Per case, a tailored analysis was presented. Markers indicative of malignancy were searched for in the tumoral region.

Overall, the most relevant image pattern associated with malignancies observed in PAM 2 images was the presence of spotty signals within the tumoral region. This can be attributed to inhomogeneities within the tumor itself, or to the small vessels around it. As shown in Fig 6D and Fig 8A–8C, this pattern was observed in cases 2 and 3 in which the main lesions were relatively large in size (25 mm and 30 mm, respectively) and located close to the skin (inferior part of the breast against the thoracic wall and behind the nipple-areola complex, respectively). Hence, their localization in the PA image was relatively easy and the areas were sufficiently illuminated. This finding is comparable to what was shown by Toi et al., who identified the presence of spotty signals inside the tumor as a marker for breast cancer. It was also stated that it might be a means of discriminating between invasive breast cancer and ductal carcinoma in situ [7]. In Ref. [8], scattered hypoxic points were hypothesized to be attributable to intratumoral bleeding.

Entropy maps, which describe the local degree of disorder in the image, showed promising results only for case 3, where we observed a relatively high level of entropy at the tumor site (Fig 8B and 8D). This finding is supported by a previous study by Lin et al. [10], which showed that cancerous regions can be characterized in a PA image by regions with high entropy. The

hypothesis behind this finding is that cancer-associated angiogenesis induces the formation of chaotic and irregular vascular networks, and one of the expected image features can be the presence of vessels showing abrupt changes in amplitude. However, in case of insufficient illumination not covering the entire breast, entropy maps lose consistency and may fail in detecting abnormal regions in dark (insufficiently illuminated) regions of the volume. This problem was experienced in three out of the four analyzed cases (data not shown). Fig 8D also partly shows this problem where in the inferior part of the breast a low level of entropy is seen. This lower entropy is attributable to insufficient light fluence at 1064 nm illumination when a comparison is made with the processed reconstructed MIP image (Fig 8C, right).

For cases 1 and 4, it was not possible to find image patterns within the tumoral region that can be linked to malignancy. For case 1, it was relatively hard to identify the coronal PA slab that contained the same tissue as the coronal MR slab with the (main) lesion. As mentioned, similarities in vessel structures and shapes were used for orientation, but differences in the shape of the pendant breast in air (MR) and water (PAM 2) complicated this process. For example, vessels present in one PA slab of 20 mm could be deformed and spread over multiple MR slabs of 20 mm, due to the longer pendant breast when hanging in air. Moreover, in Fig 4A and 4B, we see that the breast was not optimally illuminated, a problem with this system which was already known [11]. With 1064 nm illumination, less of the breast was imaged than with 755 nm. Apart from correlation and localization difficulties, it can be that the imaging depth was not sufficient to include the tumoral region. For case 4, the coronal PA MIP image (S4 Fig) contained many (superficial) blood vessels, making it difficult to find descriptors of the malignancy located close to the pectoralis muscle. Therefore, a coronal MR slab of tissue containing the lesion was compared to a PA slab of tissue at the same location. Although also in this case we do not see features relatable to the tumor itself, we hypothesize that we see small vessels, in a comb-like structure, associated with the lesion (marked as V2 in Fig 10A and 10B).

To summarize, we found unique malignancy-associated image features within the tumoral region in two out of four cases (case 2 and 3), and tumor-related vessels in two cases (case 3 and 4). But in the current research phase, we need a priori knowledge about the lesion and its location, and correlate PA image features (mostly vascular structures) with image features of conventional clinical images. For example, in a blind study, without the MR image of case 4, the structures indicated with arrowhead 2 in Fig 10A and 10B would not have been identified as suspicious and related to a malignant lesion. Other studies also evaluate their PA breast images in correlation with MR or US images, either in an overlay or by using a hybrid imaging system [7, 8]. It could be that this is only necessary in the current 'learning phase', where we are looking for and investigating the feasibility and predictive value of image descriptors indicative of malignancy, and in need of a reference. Or, perhaps the technique could one day have a supplementary role in the clinic, used in a hybrid setting with MR or US. But at least for the research phase, until we know exactly the PA appearance of breast tumors, it would be recommended to continue correlating PA images with MR images, or have a hybrid system (with US seems the most obvious right now). In this work, resemblance between PA and MR vessels structures was used to identify tumoral regions. This would be easier when deforming images of one of the modalities to obtain similar breast shapes, as was done in Ref. [7]. This was not possible with the data presented in this work, due to a lack of a sufficient number of landmarks identifiable in both modalities, especially in 'dark' regions of the image.

In addition to entropy, there are other image descriptors (textural features) that can be used to quantitatively characterize an image. We evaluated the use of other first-order statistical descriptors and advanced textural features such as the co-occurrence matrix and the grey level size zone matrix features. However, it was extremely difficult to identify a set of features able to

identify the tumoral region with good accuracy. Furthermore, the absence of an accurate localization of the tumor in the photoacoustic image limits the ability to evaluate the predictive performance of such descriptors. And, similarly as was mentioned for entropy maps, such methods would need homogeneous illumination, to be able to accurately compare malignant regions to healthy tissue. A homogenous illumination not only requires an improved illumination scheme, but also a central position of the breast within the imaging tank, as was also shown in Ref. [12] by making use of cups supporting the breast.

Other groups presented methods for identifying tumors related to vessel density. Lin et al. calculated vessel density values in the tumoral region and found it to be almost always higher than the healthy surrounding tissue [9]. Toi et al. counted all trunks and branches of vessels and found higher numbers in the affected breast than in the contralateral breast [7]. With this information alone, one can obviously not yet localize and visualize the malignant lesion. Strategies like this are promising for discriminating malignant from healthy tissue, but were unfortunately not applicable to our data, due to the inhomogeneous illumination and variations of breast positioning within the imaging volume.

This brings us to an important recommendation for further research: improve the homogeneity of the illumination and ensure central and reproducible positioning of the breast within the imaging volume, to optimize both illumination and detection. For our specific imaging system, the advantages of centrally positioning the breast have already been shown in Ref [12], where the design and development of breast-supporting cups is described. Scans obtained with these supporting cups also profit from stabilization, improving image quality. Furthermore, it would be beneficial to let the illumination beams of the two wavelengths spatially overlap, to open possibilities for oxygen saturation assessment, to enhance the method's sensitivity given the clinical relevance of hypoxia [16].

Another option to enhance sensitivity of the method might lie in the use of contrast agents. Perhaps the intrinsic contrast of tumor-related vasculature is not going to be enough in all cases, and an extrinsic form of contrast would be needed, similar to the contrast enhancement after gadolinium injection in MR imaging. Apart from enhancing the 'regular' PA contrast, exogenous contrast agents may allow imaging of cellular and molecular events [17]. Contrast agents may for example bind to molecular indicators of angiogenesis (active targeting), or particles may accumulate in the tumor due to its leaky and unorganized blood vessels (passive targeting), resulting in a contrast mechanism very similar to that of DCE-MR. A lot of research goes into this topic, for example into developing smart particles that are sensitive to the acidity of the tissue and are able to shift absorption peaks [18]. However, one of the strengths of photoacoustic imaging, especially in comparison with DCE-MR imaging, is the noninvasiveness when making use of the tissue's intrinsic contrast. Therefore, the use of an exogenous contrast agent for breast imaging should only be considered when proven necessary.

In general, it seems that we have to look for additional features, apart from the morphology of blood vessels, to be able to discriminate malignant from healthy tissue when assessing images without a priori information. Options already mentioned are to do hybrid imaging with MR or US, and quantitative image interpretation methods like vessel density maps. Oxygen saturation assessment has been shown to be of added value, in small cohorts [7, 8]. With the PAM 2 system, we have seen image features that are indicative of breast malignancies in a few patients, but more research is needed to confirm these findings and hopefully deduce more features to enhance the method's sensitivity. To develop a full diagnostic-feature set, studies will be performed on a larger dataset which will include benign cases for comparison. It is therefore recommended to continue research with an updated version of the current system, with an improved illumination system (in terms of illuminating more of the tissue surface, but also spatially overlapping of beams of different wavelengths). Moreover, a means to

stabilize and position the breast, such as described in Ref [12], is recommended for further research. And ideally, an updated system would have a built-in extra modality for anatomical reference, where ultrasonography is an obvious choice.

## Supporting information

**S1 Table. Subject characteristics.** 'NST' = non-specific type, 'DD' = differential diagnosis, 'oc' = o'clock. Subjects with IDs 1 through 4 are discussed in detail in this work. [a]Diagnosis of malignant lesions was based on histopathological examination of biopsy specimens. For cases studied in detail in this work, results of histology of lumpectomy or mastectomy were also obtained, but not included in this table.
(DOCX)

**S1 Fig. Case 1: Comparison of photoacoustic with conventional MR images.** (A) Photoacoustic maximum intensity projections (MIPs) in two planes (coronal (top) and transverse (bottom)), at two illumination wavelengths. (B) Post-contrast dynamic T1 MR MIPs in the same two planes. All scale bars represent 20 mm.
(TIF)

**S2 Fig. Case 2: Comparison of photoacoustic with conventional MR images.** (A) Photoacoustic maximum intensity projections (MIPs) in two planes (coronal (top) and transverse (bottom)), at two illumination wavelengths. (B) Post-contrast dynamic T1 MR MIPs in the same two planes. All scale bars represent 20 mm.
(TIF)

**S3 Fig. Case 3: Photoacoustic maximum intensity projections (MIPs) in two planes (coronal (top) and transverse (bottom)), at two illumination wavelengths.** All scale bars represent 20 mm.
(TIF)

**S4 Fig. Case 4: Comparison of photoacoustic with conventional MR images.** (A) Photoacoustic maximum intensity projections (MIPs) in two planes (coronal (top) and transverse (bottom)), at two illumination wavelengths. (B) Post-contrast dynamic T1 MR MIPs in the same two planes. All scale bars represent 20 mm.
(TIF)

**S5 Fig. Case 1: Photoacoustic maximum intensity projections (MIPs) in two planes (coronal (top) and transverse (bottom)), at two illumination wavelengths of the contralateral healthy breast.** All scale bars represent 20 mm.
(TIF)

**S6 Fig. Case 3: Photoacoustic maximum intensity projections (MIPs) in two planes (coronal (top) and transverse (bottom)), at two illumination wavelengths of the contralateral healthy breast.** All scale bars represent 20 mm.
(TIF)

## Acknowledgments

Authors would like to thank patients for participating in this study and all involved medical staff for their collaboration. Rutger Pompe van Meerdervoort and Laurens Alink are thanked for technical help and assistance.

## Author Contributions

**Conceptualization:** S. M. Schoustra, C. A. H. Klazen, M. van der Schaaf, W. Steenbergen, S. Manohar.

**Data curation:** S. M. Schoustra.

**Formal analysis:** S. M. Schoustra, B. De Santi, T. J. P. M. op 't Root, J. Veltman.

**Funding acquisition:** W. Steenbergen, S. Manohar.

**Investigation:** S. M. Schoustra, B. De Santi, T. J. P. M. op 't Root, J. Veltman.

**Methodology:** S. M. Schoustra, C. A. H. Klazen, M. van der Schaaf, W. Steenbergen, S. Manohar.

**Project administration:** S. M. Schoustra.

**Software:** B. De Santi, T. J. P. M. op 't Root.

**Supervision:** W. Steenbergen, S. Manohar.

**Visualization:** S. M. Schoustra, B. De Santi.

**Writing – original draft:** S. M. Schoustra.

**Writing – review & editing:** S. M. Schoustra, B. De Santi, J. Veltman, W. Steenbergen, S. Manohar.

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
