## [Decision Letter · Decision Letter 0]

6 Sep 2022

PONE-D-22-20783Imaging breast malignancies with the Twente Photoacoustic Mammoscope 2PLOS ONE

Dear Dr. Schoustra,

Thank you for submitting your manuscript to PLOS ONE. After careful consideration, we feel that it has merit but does not fully meet PLOS ONE’s publication criteria as it currently stands. Therefore, we invite you to submit a revised version of the manuscript that addresses the points raised during the review process.

We look forward to receiving your revised manuscript.

Kind regards,

Jeeun Kang, Ph.D.

Academic Editor

PLOS ONE

Journal Requirements:

"I have read the journal's policy and the authors of this manuscript have the following competing interests: author Klazen has a financial interest in PA Imaging Holding B.V., via PAMARA Holding B.V."

"I have read the journal's policy and the authors of this manuscript have the following competing interests: author Klazen has a financial interest in PA Imaging Holding B.V., via PAMARA Holding B.V."

Reviewers' comments:

Reviewer's Responses to Questions

**Comments to the Author**

1. Is the manuscript technically sound, and do the data support the conclusions?

Reviewer #1: Partly

Reviewer #2: Yes

2. Has the statistical analysis been performed appropriately and rigorously? 

Reviewer #1: N/A

Reviewer #2: Yes

3. Have the authors made all data underlying the findings in their manuscript fully available?

Reviewer #1: Yes

Reviewer #2: Yes

4. Is the manuscript presented in an intelligible fashion and written in standard English?

Reviewer #1: Yes

Reviewer #2: Yes

5. Review Comments to the Author

Reviewer #1: The authors reported clinical measurements of breast cancer patients with the 3D Twente photoacoustic mammoscope prototype. Specifically, 30 patients in total were scanned and 4 out of 19 malignant cases were closely investigated. Photoacoustic imaging results were qualitatively compared with craniocaudal X-ray images, ultrasound images, and MRI images. Even though a larger cohort is expected for more convincing conclusions, the scientific findings of the four cases are of interest to this journal. The following aspects can be improved before further consideration of publication:

1. One caveat of the study is that there is no control group. It is unclear how different images will look like in benign cases, especially in PA images. Authors should at least analyze some benign cases and show the X-ray, MRI, US, and PA images for comparison.

2. The authors claim that “to enhance signals coming from the deeper vessels, adaptive intensity modulation was applied”. However, from the justification as well as results shown in figures (such as Fig 6), it seems that adaptive intensity modulation does not compensate for low illumination in deep regions. Instead, it adjusts image intensity with respect to local variability, which is more anatomy dependent rather than depth dependent. Please elaborate or correct.

3. To compensate for depth dependent PA intensity, optical fluence compensation is necessary, which is missing in this work. There have been multiple works on optical fluence compensation for breast. Suggest authors at least implementing a simple model-based method to consider exponentially attenuated light fluence given the boundary of the breast.

4. Also in image processing, why are the final images obtained by summing the adaptive modulated image and the median filtered image, instead of applying the adaptive modulation directly on the filtered image?

5. It is difficult to correlate different modalities without registration. The authors discussed the difficulty of cross-modality registration due to a lack of anatomical landmarks, but also mentioned a previous work where MRI and PA images are deformably registered and fused in reference [7] in Introduction. Why not applying the same technique to register at least MR and PA images?

Other comments:

6. In Materials and Methods, the authors mentioned that two US probes were used in the study, i.e. an L12-5 50 and an L18-5. However, only one ultrasound image was shown for each case in Results. Please elaborate the difference of the images acquired from the two probes, and how they were used in diagnosis.

7. How does the entropy map look like for the other three cases?

8. In discussion, the authors explained that the oxygen saturation was not evaluated due to the difference in illuminated tissue with the two wavelengths. This is confusing, why would the illuminated tissue change at different wavelengths? Please clarify.

9. The discussion of improvements achieved from this work compared with the previous generation of Twente mammoscope [5, 6] is inadequate.

10. The discussion can be organized more systematically and concisely by concluding all cases by their features, instead of listing one by one again as in Results. Suggest authors using “can be” instead of “might be” or “seems that” to conject.

Reviewer #2: The manuscript verified the clinical feasibility of PAM-02 system to diagnose breast cancer. Overall the manuscript is acceptable to publication. Please address the minor question of the PAM-02 system.

- The authors used 795nm and 1064nm lasers for imaging. Please justify why both wavelength are chosen. and other research (10.1038/s41467-018-04576-z) stated single 1064 laser was used to detect breast cancer, why did you use dual-band laser?

6. PLOS authors have the option to publish the peer review history of their article (what does this mean?). If published, this will include your full peer review and any attached files.

Reviewer #1: No

Reviewer #2: No

---

## [Author Response · Author response to Decision Letter 0]

8 Nov 2022

Dear editor and dear reviewers,

responses to your comments have been provided in the Response to Reviewers uploaded document.

Best regards,

Bruno De Santi

---

## [Decision Letter · Decision Letter 1]

29 Nov 2022

PONE-D-22-20783R1Imaging breast malignancies with the Twente Photoacoustic Mammoscope 2PLOS ONE

Dear Dr. De Santi,

Thank you for submitting your manuscript to PLOS ONE. After careful consideration, we feel that minor revision would make your article fully satisfy PLOS ONE’s publication criteria. Therefore, we invite you to submit a revised version of the manuscript that addresses the minor points raised during the review process.

We look forward to receiving your revised manuscript.

Kind regards,

Jeeun Kang, Ph.D.

Academic Editor

PLOS ONE

Journal Requirements:

Reviewers' comments:

Reviewer's Responses to Questions

**Comments to the Author**

1. If the authors have adequately addressed your comments raised in a previous round of review and you feel that this manuscript is now acceptable for publication, you may indicate that here to bypass the “Comments to the Author” section, enter your conflict of interest statement in the “Confidential to Editor” section, and submit your "Accept" recommendation.

Reviewer #1: (No Response)

Reviewer #2: All comments have been addressed

2. Is the manuscript technically sound, and do the data support the conclusions?

Reviewer #1: Yes

Reviewer #2: Yes

3. Has the statistical analysis been performed appropriately and rigorously? 

Reviewer #1: Yes

Reviewer #2: Yes

4. Have the authors made all data underlying the findings in their manuscript fully available?

Reviewer #1: Yes

Reviewer #2: Yes

5. Is the manuscript presented in an intelligible fashion and written in standard English?

Reviewer #1: Yes

Reviewer #2: Yes

6. Review Comments to the Author

Reviewer #1: Partially addressed:

Comment 1: Let alone the benign cases for now, is it possible to show images of normal breasts so that readers can have a clearer sense how the vascular features are different in breast tumors from normal breasts?

Suggestions:

Comments 2, 3, 4, 5, 7 are well addressed. If the authors choose to publish the peer review history then it might be fine. Otherwise, the revised manuscript is a bit too simplified. If the authors choose not to publish the peer review history, suggest authors include the analysis and justifications that you responded to the rebuttal to the Discussion Section to make the manuscript more comprehensive. Figures R2, R3, R5, R7 should be added to Supporting Information. Figure R4 should be added to Methods.

Reviewer #2: (No Response)

7. PLOS authors have the option to publish the peer review history of their article (what does this mean?). If published, this will include your full peer review and any attached files.

Reviewer #1: No

Reviewer #2: No

---

## [Author Response · Author response to Decision Letter 1]

13 Jan 2023

We thank both reviewers for the appreciation of the manuscript and the first response to the reviewers.

We accepted the suggestions by reviewer 1. The submission is modified as follows:

1) We added Supplementary Figures 5 and 6, which show the photoacoustic reconstructions of the healthy contralateral breast for cases 1 and 3. As mentioned in our first response, we believe that at this stage of the research with our instrument, a comparison between ipsilateral and contralateral breasts is not insightful. The reason is mainly the absence of immobilization of the breast, which can result in movements in one case but less in another case. This can result in unpredictable artifacts and blurring in one breast but not in the other, making comparison between images of the two breasts unproductive. 

2) As suggested, we give permission to publish the first response to reviewers.

3) Finally, reviewer 1 also suggested including Figure R4 (in the response to reviewers document) in the Methods section of the manuscript. As this figure was sharing part of the content with Figure 2, we decided to combine the two figures in order to clearly show the pipeline steps and each intermediate result. 

Best regards,

Bruno De Santi

---

## [Editor Report · Decision Letter 2]

24 Jan 2023

Imaging breast malignancies with the Twente Photoacoustic Mammoscope 2

PONE-D-22-20783R2

Dear Dr. De Santi,

We’re pleased to inform you that your manuscript has been judged scientifically suitable for publication and will be formally accepted for publication once it meets all outstanding technical requirements.

Kind regards,

Jeeun Kang, Ph.D.

Academic Editor

PLOS ONE

---

## [Editor Report · Acceptance letter]

20 Feb 2023

PONE-D-22-20783R2 

Imaging breast malignancies with the Twente Photoacoustic Mammoscope 2 

Dear Dr. De Santi:

I'm pleased to inform you that your manuscript has been deemed suitable for publication in PLOS ONE. Congratulations! Your manuscript is now with our production department. 

Kind regards, 

on behalf of

Dr. Jeeun Kang 

Academic Editor

PLOS ONE